# The near-atomic cryoEM structure of a flexible filamentous plant virus shows homology of its coat protein with nucleoproteins of animal viruses

Xabier Agirrezabala[1], Eduardo Méndez-López[2,3], Gorka Lasso[4], M Amelia Sánchez-Pina[2,3], Miguel Aranda[2,3], Mikel Valle[1]*

[1]Structural Biology Unit, Center for Cooperative Research in Biosciences, Derio, Spain; [2]Centro de Edafología y Biología Aplicada del Segura, Murcia, Spain; [3]Consejo Superior de Investigaciones Científicas, Murcia, Spain; [4]Department of Biochemistry and Molecular Biophysics, Columbia University, New York, United States

**Abstract** Flexible filamentous viruses include economically important plant pathogens. Their viral particles contain several hundred copies of a helically arrayed coat protein (CP) protecting a (+)ssRNA. We describe here a structure at 3.9 Å resolution, from electron cryomicroscopy, of *Pepino mosaic virus* (PepMV), a representative of the genus *Potexvirus* (family *Alphaflexiviridae*). Our results allow modeling of the CP and its interactions with viral RNA. The overall fold of PepMV CP resembles that of nucleoproteins (NPs) from the genus *Phlebovirus* (family *Bunyaviridae*), a group of enveloped (-)ssRNA viruses. The main difference between potexvirus CP and phlebovirus NP is in their C-terminal extensions, which appear to determine the characteristics of the distinct multimeric assemblies – a flexuous, helical rod or a loose ribonucleoprotein. The homology suggests gene transfer between eukaryotic (+) and (-)ssRNA viruses.

*For correspondence: mvalle@cicbiogune.es

**Competing interests:** The authors declare that no competing interests exist.

## Introduction

Flexible filamentous viruses are ubiquitous plant pathogens that have an enormous impact in agriculture (*Revers and Garcia, 2015*). Their infective particles are non-enveloped and flexible rod-shaped virions (*Kendall et al., 2008*) and contain several hundreds of copies of a coat protein (CP) arranged in a helical fashion protecting a ssRNA of positive polarity, or (+)ssRNA (*Kendall et al., 2008*). They are distributed in families *Alphaflexiviridae*, *Betaflexiviridae*, *Closteroviridae*, and *Potyviridae*, which have different genomic organizations (particularly different between *Potyviridae* and the rest of the groups) and belong to different superfamilies, but it is thought that their CPs have strong evolutionary relationships (*Koonin et al., 2015*). Structural studies of virions by X-ray fiber diffraction and cryoEM have indeed revealed a common architecture for flexible plant viruses. The filaments are 120-130 Å in diameter, and the CPs are arranged following helical symmetry with slightly less than 9 subunits per turn (*Kendall et al., 2008*). This overall arrangement is shared by *Soybean mosaic virus* (SMV) a potyvirus (from the family *Potyviridae*), and three different potexviruses (family *Alphaflexiviridae*), *Potato virus X* (PVX), *Papaya mosaic virus* (PapMV), and *Narcissus mosaic virus* (NMV) (*Kendall et al., 2013*, *Yang et al., 2012*, *Kendall et al., 2008*). The flexibility of the virions has limited high-resolution structural studies, and most of the previous data were at moderate resolution. Very recently, the cryoEM structure of *Bamboo mosaic virus* (BaMV), another potexvirus, was determined at 5.6 Å (*DiMaio et al., 2015*), and the CP was modeled based on the atomic structure of a

**eLife digest** A group of "flexible filamentous" viruses can cause serious diseases in a wide variety of crops and other plants. Each virus particle contains a single molecule called ribonucleic acid (RNA), which is protected by hundreds of copies of a coat protein. The RNA and coat proteins are arranged in a helical fashion to make a flexible rod-shaped particle.

The flexibility of these viruses makes it difficult to carry out in-depth studies of their three-dimensional structures. As a result, we do not know how the RNA and coat proteins interact to form the structure of each virus particle. Agirrezabala et al. used a technique called cryo-electron microscopy (or cryoEM for short) to generate a highly detailed three-dimensional model of a flexible filamentous virus called Pepino Mosaic Virus.

Agirezabala et al.'s findings reveal how the virus particles assemble, and the interactions between the coat protein and the ssRNA. Unexpectedly, the structure of the coat protein from Pepino Mosiac Virus is very similar to the structure of "nucleoproteins" from a group of viruses called the Phleboviruses, which infect animals. This similarity is striking and suggests that the gene that encodes these proteins has been transferred between the two groups of viruses during evolution. A future challenge is to find out whether this similarity extends to other groups of viruses.

truncated version of the CP from *Papaya mosaic virus* (PapMV CP) (*Yang et al., 2012*). The work revealed that N- and C-terminal extensions of the CP mediate viral polymerization and allow for the flexuous nature of the virions.

*Pepino mosaic virus* (PepMV) is another potexvirus which has emerged recently (*Jones et al., 1980*), progressing from endemic to epidemic in tomato crops causing severe economic losses worldwide (*Hanssen and Thomma, 2010*). PepMV is transmitted by mechanical contact and virions contain a (+)ssRNA of about 6.4 kb (*Aguilar et al., 2002*). The PepMV CP is strictly required for cell-to-cell movement of the virus (*Sempere et al., 2011*).

We present the cryoEM structure of PepMV virions at 3.9 Å of resolution. The near-atomic three-dimensional (3D) map allows for accurate modeling of the CP, the viral RNA, and their interaction. In vivo functional studies of several CP mutants confirm the role of several residues in RNA binding and polymerization. Surprisingly, we have also found a clear structural homology between the CP of flexuous viruses and the NP of the genus *Phlebovirus*, a group of enveloped viruses with a segmented (-)ssRNA genome. The NPs from phleboviruses are associated with the viral genome in loose ribonucleoproteins (RNPs) (*Raymond et al., 2012*) protected inside an envelope, in which inserted glycoproteins construct an icosahedral shell (*Huiskonen et al., 2009*, *Freiberg et al., 2008*). Despite the divergence of both viral groups, CP from potexviruses and NP from phleboviruses have the same all α-helix fold, and their high similarity suggests a horizontal gene transfer event between these evolutionary distant groups of eukaryotic RNA viruses.

## Results and discussion

### CryoEM structure of PepMV

We have analyzed by cryoEM PepMV virions isolated from infected *Nicotiana benthamiana* plants (*Figure 1—figure supplement 1*). The 3D map at 3.9 Å of resolution (*Figures 1A,B* and *Figure 1—figure supplement 3*) was calculated by single particle-based helical image processing implemented in Spring software (*Desfosses et al., 2014*). The cryoEM map reveals a left-handed helix with a diameter of 130 Å and an inner narrow channel of 13 Å, and the structure shows a pitch of 34.6 Å and 8.7 CP copies per turn. This overall helical arrangement is in agreement with previous works with other potexviruses (*Kendall et al., 2013*, *Yang et al., 2012*), but in the current case and in the recent cryoEM structure of BaMV (*DiMaio et al., 2015*) the attained resolutions allowed a clear assignment of the symmetry. Our cryoEM map reaches near-atomic resolution, to date the highest resolution structural data for a flexible filamentous virion, where bulky protein side chains are discernible (*Figure 1C*). This allowed us to generate an atomic model for PepMV CP (*Figure 1D*) by iterative modeling starting with the atomic structure of PapMV CP (see 'Materials and methods').

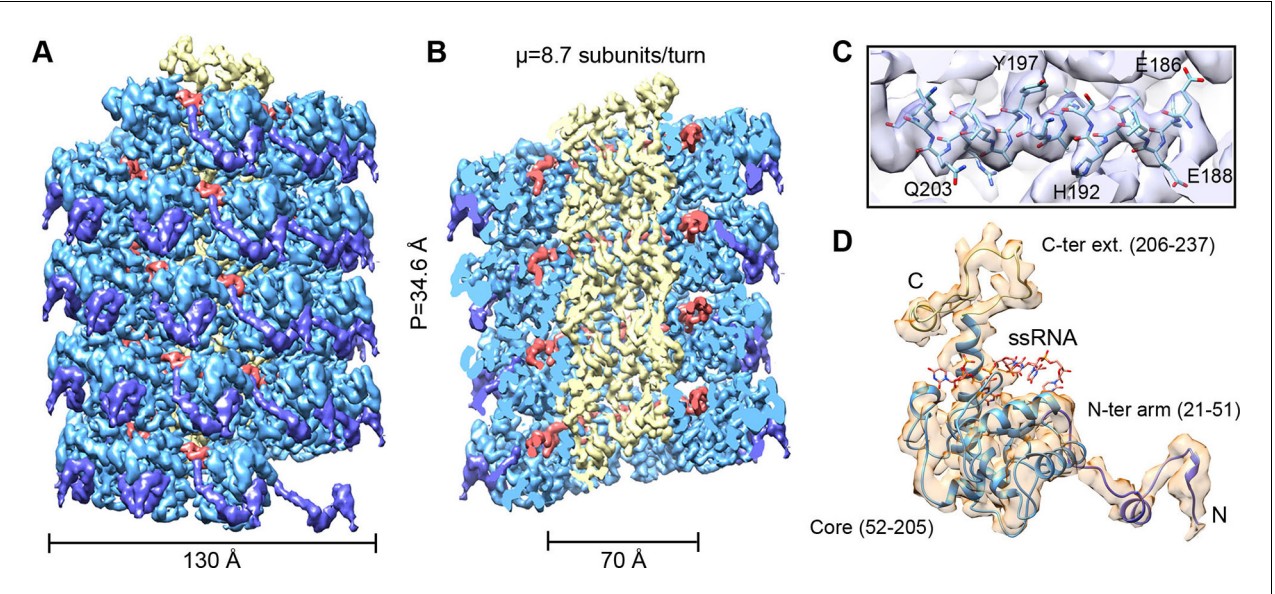

**Figure 1.** CryoEM structure of PepMV and atomic model for its CP. (**A, B**) Renderings of the 3D density map for PepMV that displays a left-handed helical symmetry with 34.6 Å of helical pitch (P). The map is seen segmented domain-wise. The cut-away view (**B**) reveals the location of the ssRNA (red). (**C**) Close-up view of a region from the cryoEM map rendered in semi-transparent mode, together with the atomic model calculated for PepMV CP. (**D**) Isolated density for a PepMV CP subunit shown semi-transparent, and representation of the PepMVCP atomic model. Color code for PepMV CP domains: core region, blue; N-terminal arm, purple; and C-terminal extension, yellow. CP, coat protein, PepMV, *Pepino mosaic virus*, PepMV CP, *Pepino mosaic virus* coat protein.

The following figure supplements are available for figure 1:

**Figure supplement 1.** Electron micrograph of PepMV cryoEM data.

**Figure supplement 2.** Helical symmetry search.

**Figure supplement 3.** Estimation of resolution for the cryoEM map of PepMV virions.

**Figure supplement 4.** Local resolution measurement in isolated PepMV CP subunit.

**Figure supplement 5.** Comparison between modeled PepMV CP and the atomic structure of PapMV CP.

**Figure supplement 6.** Table of some figures of merit for the structure of the modeled PepMV CP.

The PepMV CP structure thus generated has three major regions: the core; an N-terminal flexible arm; and a C-terminal extension. The first 20 amino acids in the N-terminal side are not included in the atomic model because the region projects outwards, and its density vanishes due to high flexibility. The modeled structure for PepMV CP is, as defined for PapMV CP, an all-helix fold, and the low RMSD between the two structures (1.5 Å in the core region) reflects their clear homology (*Figure 1—figure supplement 5*).

## CP and ssRNA interaction

The high level of structural details of our cryoEM map permits the segmentation of the density for the ssRNA and the analysis of the protein–RNA interactions. The ssRNA runs in a helix of 70 Å in diameter (*Figure 1B*). Each CP binds five ribonucleotides (*Figure 1D*), so that the entire genome of PepMV would require 1290 copies of PepMV CP spanning 510 nm (3.95 Å of axial rise/subunit), which agrees with the 509 nm length originally reported for the virus (*Jones et al., 1980*). The signal clearly separates the individual nucleotides, but we could not identify the bases due to the helical averaging of the variable sequence of the ssRNA. In order to explore the protein–ssRNA

interactions, we modeled a polyU and included a set of four PepMV CP monomers in a molecular dynamics flexible fitting (MDFF) simulation (*Trabuco et al., 2008*) (*Figure 2—figure supplement 1* and *2*). The results show that the RNA resides in a continuous groove with a high electropositive potential (*Figure 2A*) constructed by CPs from consecutive turns of the helix. Local resolution measurements in the cryoEM map suggest a high variability in positions 3 and 4 at the segmented density for the ssRNA (*Figure 2B*). Several amino acids display potential interactions with the phosphate backbone of the ssRNA. S92 and S94 in one face, and Q203 at the opposite side of the ssRNA might interact at positions 1 and 3 (*Figure 2C*). Backbone torsion angles of 85° and 130° at these positions allow the binding of the base at position 2 into a deep pocket (*Figure 2D*). Three polar-charged residues, R124, D163, and K196 might establish H-bonding contacts with this RNA base. D163 is in the most conserved region within the genus *Potexvirus*, and the consensus sequence FDFFD (*Figure 2—figure supplement 3*) constructs the floor of the RNA binding pocket. This pocket is large enough to accommodate pyrimidines or purines, and the presence of three amino acids ensures the interaction with the RNA regardless of the nucleotide..

We tested several PepMV CP mutants in trans-complementation assays (*Sempere et al., 2011*). Here, a PepMV construct that expresses GFP instead of CP (PepGFPΔCP) acts as reporter of the complementation by the co-expression of selected CP mutants. The fluorescent signal by GFP allowed following the cell-to-cell movement of the virus. The results (*Figure 2E*) reveal that the three amino acids at the binding pocket for nucleotide at position 2 (R124, D163 and K196) are required to complement the cell-to-cell movement of the CP-defective PepMV mutant, suggesting a disruption of the CP-RNA interaction needed in virus intercellular transport. Mutant K93A also impairs viral propagation within the inoculated leaf, probably due to structural changes at the loop that contains S92 and S94. In the case of mutant Q203A, it seems that there is a reduced movement between cells but not full impairment.

We also tested the production of fully assembled virions analyzing by Western blot virion preparations from infected leaves (see 'Materials and methods'). Attempts to purify PepMV virions were only fruitful for the wt and for the Q203A mutant (*Figure 2F*), suggesting the lack of full viral assembly in the rest of PepMV CP mutants. Single amino acid changes appear to impair CP-RNA binding, probably including a cooperative effect between the hundreds of CP copies that build the virions.

## CP–CP interaction in the assembled virion

As described for BaMV (*DiMaio et al., 2015*), the helical assembly of PepMV is mediated by the CP N-terminal arm and C-terminal extension. However, the improved resolution of the current data allowed for additional insights. The N-terminal arm from the $N_i$ subunit interacts with a hydrophobic groove of the $N_{i-1}$ subunit and establishes the main side-by-side contact in the helical arrangement (*Figures 1A* and *3A*). A hydrophobic pocket in $N_{i-1}$ accommodates part of the N-terminal arm, where F28 from $N_i$ fits (*Figure 3B*). A similar type of interaction was observed between subunits in the crystal structure of PapMV CP via amino acid F13 (*Yang et al., 2012*), and for BaMV, by residue W41 (*DiMaio et al., 2015*). We tested in vivo the relevance of F28 in the assembly of PepMV by assaying a PepMVCP F28A mutant in trans-complementation and viral purification experiments. The F28 mutant allows cell-to-cell movement (*Figure 3C*) but does not produce fully assembled virions (*Figure 3D*), suggesting that the tested mutation allows for the functional separation of the cell-to-cell movement of PepMV from the encapsidation of fully assembled virions, reinforcing the notion of a non-virion RNP structure during the intercellular movement of potexviruses through plasmodesmata (*Lough et al., 2000*).

On the other hand, the C-terminal extension, a long coil with a short α-helix in its last part, builds the inner wall of the virus (*Figure 1B*). This C-terminal segment describes a turn that runs along the axial channel and creates a network of small and local interactions (*Figure 3E*), where each subunit contacts neighbours at upper and lower levels. Thus, the C-terminal extensions are responsible for the axial interactions that support helix formation. As for BaMV (*DiMaio et al., 2015*), the flexible links between PepMV CPs via N-terminal arm and C-terminal region allow for relative movements between CPs and explain the flexuous nature of the virions.

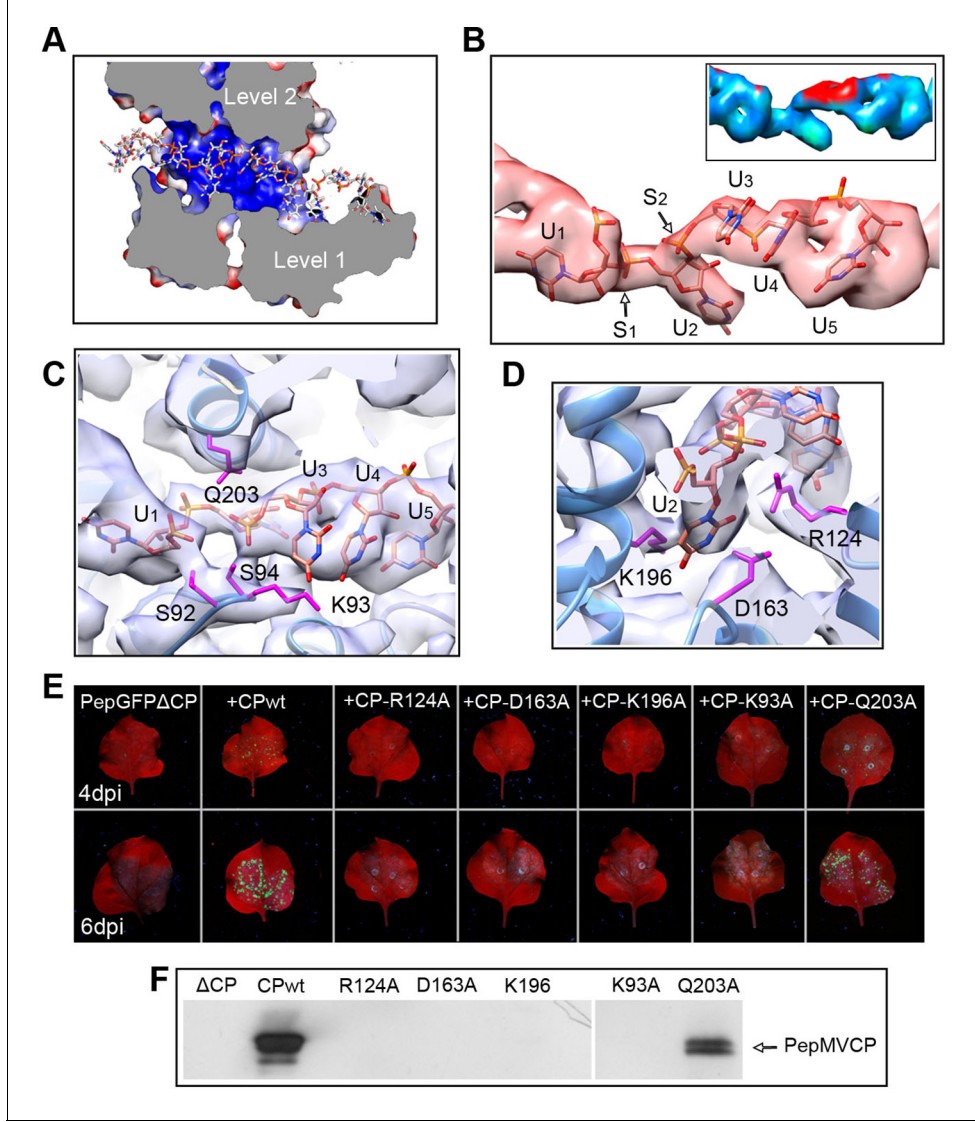

**Figure 2.** Interaction between PepMV CP and ssRNA. (**A**) Cut-away rendering of four PepMV CP subunits at consecutive turns of the helix with the ssRNA between them. Molecular surfaces are colored according to their electrostatic potential using a color scale that ranges from -5KT (red) to +5KT (blue). (**B**) Structural detail for the density of the ssRNA associated to a single PepMV CP. ssRNA is a polyU model with five nucleotides. S1 and S2 indicate switches along the phosphate backbone. The inset renders the density region for ssRNA according to local resolution measurements (blue around 4 Å, and red at 6 Å of resolution; full color scale in *Figure 1—figure supplement 4*). (**C, D**) Focus on local protein–RNA interfaces where amino acids with possible interaction with the RNA are highlighted. (**E**) Trans-complementation assays between PepGFPΔCP and several CP mutants. The images show agroinfiltrated *N. benthamiana* leaves imaged under UV light to detect the expression of GFP. Data were recorded at 4 and 6 days post inoculation (dpi). (**F**) Western blot analysis of the presence of PepMV CP in preparations of fully assembled virions from trans-complementation assays. PepMV CP, *Pepino mosaic virus* coat protein.

The following figure supplements are available for figure 2:

**Figure supplement 1.** Rendering of the atomic model construction used during MDFF for the analysis of the protein-RNA interfaces.

**Figure supplement 2.** Progress during the MDFF run.

**Figure supplement 3.** Sequence alignment between CP from several representatives of the genus *Potexvirus*.

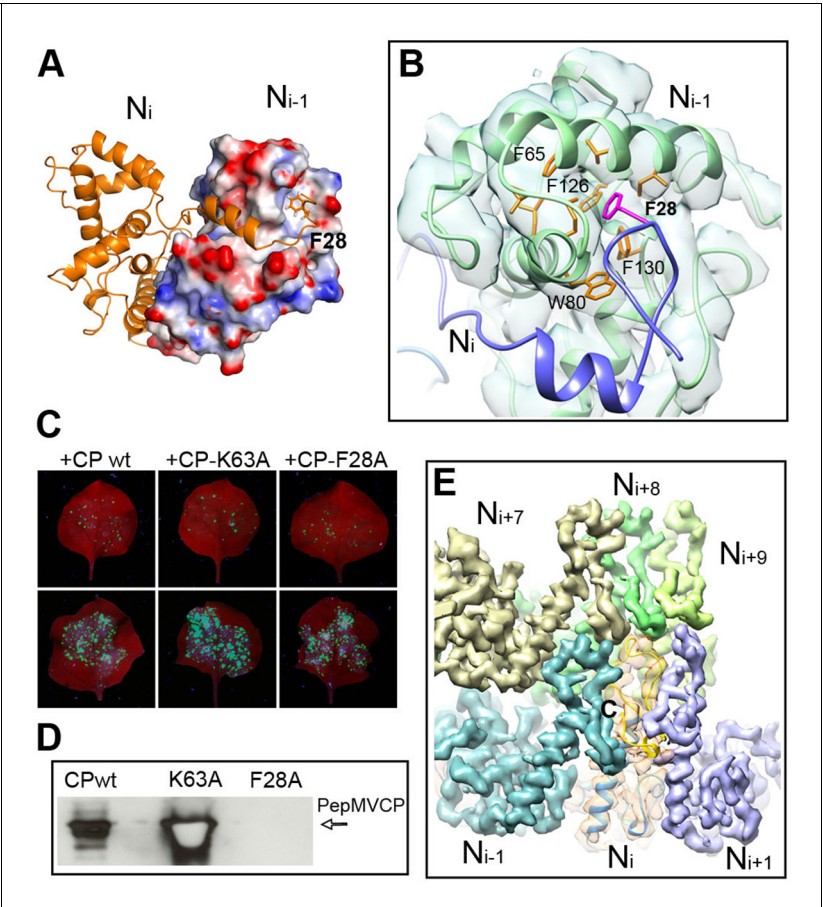

**Figure 3.** Interactions through N- and C-terminal flexible regions mediate PepMV assembly. (A) $N_i$ subunit links to a hydrophobic groove in the $N_{i-1}$ subunit via the N-terminal arm. (B) In the $N_{i-1}$ subunit a pocket of hydrophobic residues allocates F28 from the $N_i$ adjacent subunit. (C) Trans-complementation assays show that F28A mutant allows for cell-to-cell movement. (D) The analysis by Western blot of virion preparations show no signal for fully assembled virions in the trans-complementation with F28A mutant. PepMVCP mutant K63A is a positive control. (E) Six segmented densities for PepMVCP are seen from the inner side of the virion. The subunit $N_i$ is depicted semi-transparent and includes a ribbon representation for the atomic model for PepMV CP. PepMV CP, *Pepino mosaic virus* coat protein.

## Structural homology between CP from potexviruses and NP from phleboviruses

All the high-resolution structures for flexuous filamentous plant viruses correspond to representatives of the genus *Potexvirus* (family *Alphaflexiviridae*) and include our current data and previously reported structures of PapMV CP (*Yang et al., 2012*) and BaMV virions (*DiMaio et al., 2015*). We cannot confirm whether members of other families of flexuous filamentous viruses (*Betaflexiviridae*, *Closteroviridae* and *Potyviridae*) have a similar fold in their CPs. We have, nevertheless, looked for structural neighbours within the database of Dali server (*Holm and Rosenstrom, 2010*). The search revealed that the CP structure from potexviruses shares its topology with the NP of several representatives of genus *Phlebovirus* (family *Bunyaviridae*), such as *Rift Valley fever virus* (RVFV), *Toscana virus*, and *Severe fever with thrombocytopenia syndrome virus* (SFTSV). These viruses infect animals, including humans, are transmitted by arthropod vectors, and pose serious public health concerns. Genera *Potexvirus* and *Phlebovirus* belong to different superfamilies of RNA viruses when these are classified based on phylogenetic relationships among RNA-dependent RNA polymerases (RdRp) and also have different genomic organizations (*Koonin et al., 2015*). Moreover, the organization of phlebovirus particles is very different. They have segmented (-)ssRNA associated with the NP in loose RNPs (*Raymond et al., 2012*), and these are protected inside an icosahedral shell of glycoproteins inserted in a membrane, as has been observed for RVFV (*Huiskonen et al., 2009*, *Freiberg et al.,*

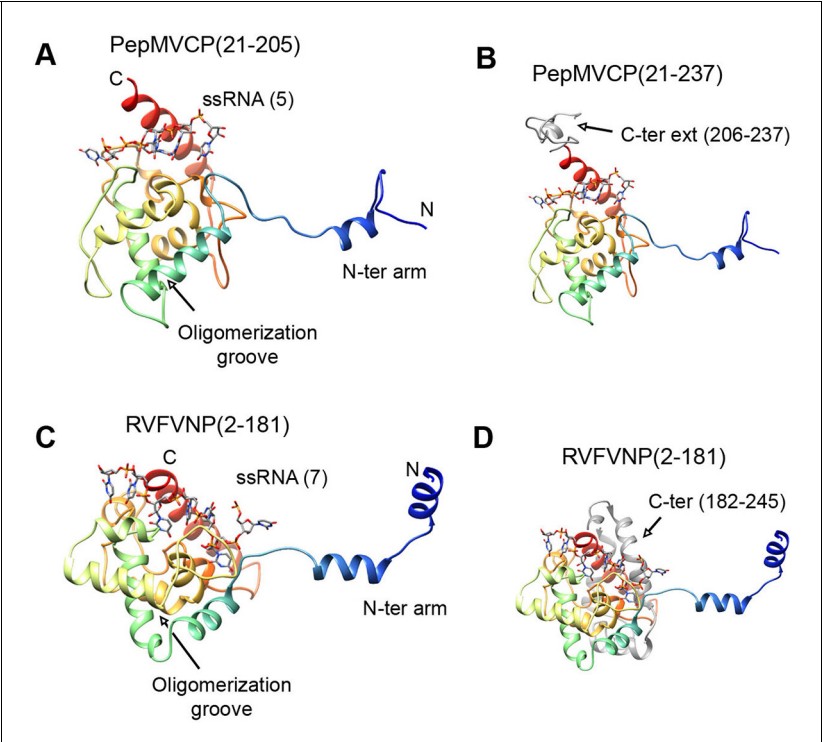

**Figure 4.** Structural homology between PepMV CP and NP from phleboviruses. (**A–D**) The atomic structures for the modeled PepMV CP and for the NP from RVFV (pdb code 4H5O (**Raymond et al., 2012**) are depicted in similar orientations. The representations include the respective ssRNAs. Both proteins are colored in rainbow mode and their similar topology is clear when their C-terminal regions are removed (**A** and **C**). Their C-termini are seen in grey color for comparison (**B** and **D**). PepMV CP, *Pepino mosaic virus* coat protein; NP, nucleoprotein.

*2008*). The topology of CP from potexviruses and of the NP from phleboviruses (illustrated for RVFV) are, nonetheless, very similar (*Figures 4A and 4C*) and show a TM-score of 0.51 when aligned, pointing to a common family fold (*Zhang and Skolnick, 2005*). The similarities include: the overall topology of the all-alpha helical domain; the N-terminal arm and its binding site from the adjacent protomer, and hence, the side-by-side mechanism for oligomerization; the groove for the ssRNA binding; and the relative positions of all these elements. The crystallographic studies with phlebovirus NPs have shown tetrameric, pentameric, and hexameric oligomers, in which each NP interacts with the core of the adjacent subunit via an N-terminal arm (*Zhou et al., 2013*) following the same side-by-side polymerization mechanism shown in potexviruses. The divergence between potexvirus CP and phlebovirus NP is in their C-terminal regions. The C-terminus in PepMV CP protrudes from the core domain (*Figure 4B*) and allows longitudinal interactions for helix building. The C-terminus of the phlebovirus NP folds back into the core region and participates in the RNA-binding site that incorporates seven bases/subunit (*Figure 4D*). The location of the C-terminal arm in phlebovirus NPs cannot support a longitudinal assembly, and hence, their RNPs remain more loosely associated (*Raymond et al., 2012*) than in their plant-virus counterparts.

Both potexvirus CPs and phlebovirus NPs have in common that they bind and protect the genomic viral ssRNA. In other regards, however, these two groups are different and evolutionary distant, no one would have anticipated the structural homology that we have found. The current work provides data that suggest a horizontal gene transfer event between genera *Potexvirus* and *Phlebovirus*, or evolutionarily related forms. While potexvirus and the rest of flexible filamentous viruses infect plants, members of the genus *Phlebovirus* infect animals. Some representatives of the family *Bunyaviridae, toposviruses*, are transmitted by and replicate in arthropods, and infect plants. A similar type of virus could have mediated the transfer of CP/NP genes between distant groups of viruses with different host range. Whether the rest of flexible filamentous plant viruses share these genes remains an open question.

## Materials and methods

### PepMV inoculation and purification

Virus preparations were obtained from *N. benthamiana* plants infected by PepMV-Sp13 (*Aguilar et al., 2002*). For inoculations, carborundum-dusted leaves of 3 weeks old *N. benthamiana* plants were rubbed with a homogenate consisting of dried material from PepMV-Sp13 infected plants ground in 30 mM sodium phosphate pH 8.0. Inoculated plants were kept in a growth chamber (16 hr photoperiod, 18°C/26°C night/day, respectively) for 2 additional weeks. Virions were purified from *N. benthamiana* systemically infected leaves following a previously described method (*AbouHaidar et al., 1998*) slightly modified. Briefly, infected leaves were homogenized in a buffer containing 0.1 M Tris-citric acid (pH 8.0), 0.2% 2-mercaptoethanol and 0.01 M sodium thioglycolate. Triton-X-100 was added (1% v/v) to the homogenized tissue and mixed for 15 min with constant stirring at 4°C. Chloroform was added to a final concentration of 25% and mixed for 30 min with constant stirring at 4°C. Then, the mixture was centrifuged for 15 min at 12,000 $g$, and viruses were precipitated from the aqueous phase by adding PEG 6000 to a concentration of 5% (w/v). The mixture was then kept with constant stirring for 60 min at 4°C, and centrifuged for 15 min at 10,000 $g$. Virus concentration was estimated by OD readings at 260 nm, with ε0.1% = 2.9 as extinction coefficient (*AbouHaidar et al., 1998*). Virions were kept refrigerated (4–6°C) in suspension buffer (0.1 M Tris-citric acid pH 8) until observation.

### Construction and assay of PepMV CP mutants

The PepMVCP gene was cloned into the binary vector pGWB2 giving raise to plasmid pGWB2-CP, and mutants were constructed based on this plasmid using standard overlapping PCR and molecular cloning methods (*Sambrook and Russell, 2001*). For functional analysis, we tested the complementation of a PepMV construct (pBPepGFPΔCP) that expressed GFP instead of CP by *trans* CP expression of selected mutants (*Sempere et al., 2011*). *Agrobacterium tumefaciens* strain C58C1 was transformed with the different CP constructs and used in trans-complementation assays. *A. tumefaciens* overnight cultures (150 ml) were centrifuged at 2500 $g$ for 10 min, and pellets were resuspended in agroinfiltration buffer (10 mM MES pH 5.5, 10 mM MgCl$_2$, and 100 μM acetosyringone) until OD$_{600}$ = 0.6. The resulting suspensions, carrying plasmids pBPepGFPΔCP (*Sempere et al., 2011*), the CP constructs and pBP19, were mixed in a 3:3:2 ratio and 6 weeks old *N. benthamiana* plants were vacuum-infiltrated with the mixture. At 4 and 6 days post infiltration, plants were observed under UV light (365 nm) using a handheld lamp (Blak Ray B100-AP lamp, UV products, Upland, CA 91786, USA). CP was detected by Western blotting in virion preparations and in protein extracts from leaves agroinfiltrated with CP constructs and pBP19. For protein extractions, 100 mg samples of *N. benthamiana* leaves were ground in 200 μl of protein extraction buffer (0.1 M Tris, pH 8.0, 0.125 mM 2-mercaptoethanol, 200 μM PMSF, 10% glycerol). Crude extracts were mixed with 5x loading buffer and separated by SDS–PAGE followed by staining with Coomassie Brilliant Blue or by electrotransfer to nitrocellulose membranes. Blots were probed with polyclonal antibodies raised in rabbits against PepMV CP (AC Diagnostics, Fayetteville, AR) followed by detection with anti-rabbit immunoglobulin G (IgG) coupled to horseradish peroxidase (Promega, Fitchburg, WI) and chemiluminescence (SuperSignal West Pico Chemiluminescent Substrate, Thermo Scientific). At 6–8 days post infiltration, *N. benthamiana* plants were harvested and the tissue was used to purify PepMV virions as described above.

### CryoEM and image processing

The PepMV samples were applied to Quantifoil R 2/2 holey carbon grids previously coated with a thin layer of carbon. Grids vitrified in a FEI Vitrobot were then transferred to a Titan Krios (FEI) electron microscope that was operated at 300 kV. Images were acquired using a Falcon II detector at nominal magnification of 59,000 and calibrated magnification of 102,967 (1.36 Å/pixel). Movie frames from the detector were recorded at a speed of 17 frames/s during 3 s. The total specimen dose was ~50 e$^-$/Å$^2$ along 51 frames. Beam-induced motion correction was performed at the level of micrographs (*Li et al., 2013*) in frames range 2-27, resulting an accumulative electron dose in the sample of the corrected images of ~25 e$^-$/Å$^2$. Contrast transfer function parameters were estimated using CTFTILT (*Mindell and Grigorieff, 2003*). Selection of helices was performed using EMAN2

(*Tang et al., 2007*). The resulting data set included 833 selected helices that were processed in SPRING (*Desfosses et al., 2014*) software-package following a strategy of single-particle based helical reconstruction. Images were CTF corrected by phase-flipping. Global and local search of optimum helical symmetry parameters (*Figure 1—figure supplement 2*) resulted in helical rise/rotation of 3.95 Å/41.1° per subunit (8.76 subunits/turn and 34.6 Å of helical pitch). The cryoEM images of helices were excised in overlapping segments of 218 Å length. The segmentation of helices was performed using several different step sizes (from 8 to 40 Å), yielding similar results. The final cryoEM map for PepMV contains information from about 170,000 asymmetric units. The resolution was estimated using the Fourier Shell Correlation (FSC) calculated between fully independent half-sets (the so-called 'gold standard') and 0.5/0.143 cutoffs in the FSC correspond to 4.5/3.9 Å resolution (*Figure 1—figure supplement 3*). Local resolution variability was also estimated using ResMap (*Kucukelbir et al., 2014*) (*Figure 1—figure supplement 4*). The cryoEM map was subjected to an enhancement of high frequencies applying a B-factor of -200 1/$Å^2$ and was low-pass filtered to 3.9 Å.

## Atomic model building

The initial atomic model for PepMV CP was generated via iTasser (*Zhang, 2008*) starting with the structure of the CP from PapMV (*Yang et al., 2012*). Segmentation of the cryoEM map and the initial rigid body fitting of the iTasser model's fragment 50-196 was done manually in Chimera (*Pettersen et al., 2004*). The sequence was set on register, and the rest of the structure built using Coot (*Emsley et al., 2010*). The model was improved by iterative cycles of manual model rebuilding. Refmac5 (*Murshudov et al., 1999*) was used to refine the model and to correct geometry/stereochemistry problems. Non-crystallographic symmetry was used in order to improve interfaces and minimize clashes between adjacent subunits. The MolProbity and clash score statistics (*Chen et al., 2010*) were in the top 100th percentile when compared with atomic structures at similar resolution (*Figure 1—figure supplement 6*). The geometry of the ssRNA (modeled as a polyU) was further improved using the Rossetta Erraser tool (*Chou et al., 2013*).

Molecular dynamics simulations were carried out with NAMD 2.9 (*Phillips et al., 2005*) through the MDFF plug-in (*Trabuco et al., 2008*). Simulations were run with the CHARMM27 force field with CMAP corrections (*Mackerell et al., 2004*) in explicit solvent at a gscale of 0.3. Simulation parameters were kept as specified by the MDFF plug-in with the exception of margin (2), cutoff (12), switchdist (10), pairlistdist (16), nonbondedFrequency (1), and fullElectFrequency (1). Simulation used restraints for secondary structure, chirality and cispeptide derived from the initial atomic model. During the first 10 ns of the simulation RNA atoms were first coupled to the density while keeping the protein backbone atoms constrained and vice versa. Following this, RNA and protein heavy atoms were simultaneously fitted into the density for 30 ns. Finally, 10,000 steps of energy minimization were performed with a grid scaling of 0 in order to increase the stability of the resulting structure. Electrostatic surface potential was calculated in Delphi (*Li et al., 2012*).

## Accession numbers

The 3D cryoEM map for PepMV and the derived CP atomic model are deposited in the Electron Microscopy Data Bank (www.emdatabank.org) and the Protein Data Bank (www.rcsb.org) under accession codes EMD-3236 and 5FN1.

## Acknowledgements

This work was supported by grants from the Spanish Ministry of Economy and Competitiveness (BFU2012-34873 and AGL2012-37390). FEM was the recipient of a FPI predoctoral fellowship from the Spanish Ministry of Economy and Competitiveness. We thank the Electron Microscopy Facility from the Laboratory of Molecular Biology (LMB-MRC, Cambridge) for the access to the electron microscope, Shaoxia Chen, Christos Savva, and Luis Rodríguez-Moreno for technical assistance.

## Additional information

### Funding

| Funder | Grant reference number | Author |
|---|---|---|
| Ministry of Economy and Competitiveness | BFU2012-34873 | Mikel Valle |
| Ministry of Economy and Competitiveness | AGL2012-37390 | Miguel Aranda |

The funders had no role in study design, data collection and interpretation, or the decision to submit the work for publication.

### Author contributions

XA, MV, Conception and design, Acquisition of data, Analysis and interpretation of data, Drafting or revising the article; EML, GL, Acquisition of data, Analysis and interpretation of data, Drafting or revising the article; MASP, Analysis and interpretation of data, Drafting or revising the article; MA, Conception and design, Analysis and interpretation of data, Drafting or revising the article

## Additional files

### Major datasets

The following datasets were generated:

| Author(s) | Year | Dataset title | Dataset URL | Database, license, and accessibility information |
|---|---|---|---|---|
| Agirrezabala X, Mendez-Lopez E, Lasso G, Sanchez-Pina MA, Aranda MA, Valle M | 2015 | Electron cryo-microscopy of filamentous flexible virus PepMV (Pepino Mosaic Virus) | http://www.ebi.ac.uk/pdbe/entry/emdb/EMD-3236 | Publicly available at the Electron Microscopy Data Bank (accession no. EMD-3236) |
| Agirrezabala X, Mendez-Lopez E, Lasso G, Sanchez-Pina MA, Aranda MA, Valle M | 2015 | Electron cryo-microscopy of filamentous flexible virus PepMV (Pepino Mosaic Virus) | http://www.rcsb.org/pdb/explore/explore.do?structureId=5FN1 | Publicly available at the Protien Data Bank (accession no. 5FN1) |

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
