## [Decision Letter]

Thank you for submitting your work entitled "The near-atomic cryoEM structure of *Pepino mosaic virus* reveals gene transfer between eukaryotic ssRNA viruses" for consideration by *eLife*. Your article has been favorably evaluated by John Kuriyan (Senior Editor) and three reviewers, one of whom, Stephen Harrison, is a member of our Board of Reviewing Editors. One of the other two reviewers, Ed Egelman, has agreed to share his identity.

The reviewers have discussed the reviews with one another and the Reviewing editor has drafted this decision to help you prepare a revised submission.

Summary:

This paper reports a structure for *Pepino mosaic virus*, a flexuous rod, like a number of other plant viruses. The structure, determined by electron cryomicroscopy, is at sufficiently high resolution to permit a proper chain trace of the capsid subunit and a clear view of the interaction of five nucleotides with each subunit. The most interesting result is the relationship of the capsid subunit to the nucleoprotein of bunyaviruses – another nice example of close relationships between plant and animal viruses, but in this case particularly striking because *Pepino mosaic virus* is a plus-strand ssRNA virus and the bunyaviruses are minus-strand viruses.

The paper is appropriate for *eLife*, but it needs extensive rewriting and some substantive revision, as outlined below. All these required changes should be straightforward.

Essential revisions:

1) More explicit comparison with BaMV. The authors should compare their atomic model with the BaMV model (5A2T.pdb). They should also include the BaMV sequence in Figure 2—figure supplement 4.

2) Figure 1—figure supplement 2: The point of this figure is not clear. It shows a helical symmetry search with a peak at 9.28 units/turn, which looks even stronger than the peak at 8.75 (the correct symmetry). Did the authors choose the latter simply because of the BaMV symmetry or because the alternative did not give a plausible map? Please clarify.

3) Figure 1—figure supplement 4. The reviewers have concerns about using ResMap. In the ResMap manual, one reads: "Helical particles are not well supported". Is this why the authors used it on an "isolated" subunit rather than on the full helix? But cutting out a subunit introduces a strong masking function, and ResMap actually works by comparing the density fluctuations within the subunit with those outside it. But the latter are now masked off. So in the figure there is a helix stated to be at ~3.5 Å resolution, but no helical grooves can be seen. The authors should show an FSC curve between that reconstruction and their atomic model.

4) In the subsection “CP and ssRNA interaction”: Interpretation of the CP-complementation studies (e.g., "suggesting a disruption of the CP-RNA interaction") seems to be flawed. Previous studies showed CP mutants that do not interrupt CP-RNA interactions and can still assemble virions (e.g., a C-terminal trucation) are defective in cell-cell movement. CP mutants that do not assemble into virions can complement the defect of cell-to-cell movement. CP mutants that do not assemble into virions can complement the defect in cell-cell movement.

5) In the last paragraph of the Introduction and in the first paragraph of the subsection “Structural homology between CP from potexviruses and NP from phleboviruses”: The icosahedral shell in RVFV is not a "capsid", which refers instead (in this case) to the helical structure associated with the RNA inside the particle. It is the "icosahedral shell of glycoproteins".

6) The title should be changed. The data do not "reveal gene transfer" – such a revelation could come only from an observation of a transfer intermediate. We suggest: The structure of a flexuous-helix plant virus shows homology of its capsid protein with the nucleoprotein of animal bunyaviruses.

7) The Abstract should be revised along the lines suggested below:

Flexible filamentous viruses include economically important plant pathogens. Their viral particles contain several hundred copies of a helically arrayed coat protein (CP) protecting a (+)ssRNA. We describe here a structure at 3.9 Å resolution, from electron cryomicroscopy, of Pepino mosaic virus (PepMV), a representative of the genus Potexvirus (family Alphaflexiviridae). Our results allow modeling of the CP and its interactions with viral RNA. The overall fold of PepMV CP resembles that of nucleoprotein (NPs) from the genus Phlebovirus (family Bunyaviridae), a group of enveloped (-)ssRNA viruses. The main difference between potexvirus CP and phlebovirus NP is in their C-terminal extensions, which appear to determine the characteristics of the distinct multimeric assemblies – a flexuous, helical rod or a loose ribonucleoprotein (RNP). The homology suggests gene transfer between eukaryotic (+) and (-) SSRNA viruses.

---

## [Author Response]

*1) More explicit comparison with BaMV. The authors should compare their atomic model with the BaMV model (5A2T.pdb). They should also include the BaMV sequence in Figure 2—figure supplement 4.*

The manuscript now contains a comparison between our model for PepMV CP and the reported BaMV CP (5A2T). This is shown in Figure 1—figure supplement 5. The figure and its legend include information about the RMSD between both atomic models and a difference in their secondary structures in one region involved in RNA binding. Also, the BaMV CP sequence has now been included in the alignment shown in Figure 2—figure supplement 3.

2) Figure 1

*—figure supplement 2: The point of this figure is not clear. It shows a helical symmetry search with a peak at 9.28 units/turn, which looks even stronger than the peak at 8.75 (the correct symmetry). Did the authors choose the latter simply because of the BaMV symmetry or because the alternative did not give a plausible map? Please clarify.*

In the search for helical symmetry we tested all the local maxima, and selected the "best looking" reconstruction, essentially the map with discernible subunits. Then we proceed with local searches around the chosen symmetry. Of course, this is not very objective, and only the high resolution of the final map confirmed the accuracy of the parameters, and there was no need to calculate maps with alternative symmetries. It is clear that the figure legend was misleading and now it has been changed.

When the map was calculated, the structure for BaMV was not available. In any case, since small structural variations in the CP might change the helical symmetry we must search the parameters in every new sample.

*3) Figure 1—figure supplement 4. The reviewers have concerns about using ResMap. In the ResMap manual, one reads: "Helical particles are not well supported". Is this why the authors used it on an "isolated" subunit rather than on the full helix? But cutting out a subunit introduces a strong masking function, and ResMap actually works by comparing the density fluctuations within the subunit with those outside it. But the latter are now masked off. So in the figure there is a helix stated to be at ~3.5 Å*

*resolution, but no helical grooves can be seen. The authors should show an FSC curve between that reconstruction and their atomic model.*

First, we calculated the local resolution using the original, unmasked and unfiltered total volume, but we displayed the output in a single, isolated CP subunit for clarity. This issue has been clarified in the corresponding figure legend.

On the other hand, in the Resmap manual, it is stated that "Helical particles are not well supported". This statement is related to the fact that Resmap needs the input volume to have the reconstructed particle well centered in the box, and long helical structures might not be suitable. We calculated the resolution using a short, well centered helical particle with a few turns of the helix, so that the requirement for a central position could be fulfilled. Calculations using Resmap have been used in recent works with helical particles. For instance in:

Gutsche et al. Near-atomic cryo-EM structure of the helical measles virus nucleocapsid. Science. 2015 May 8;348(6235):704-7;

Clare et al. Novel Inter-Subunit Contacts in Barley Stripe Mosaic Virus Revealed by Cryo-Electron Microscopy. Structure. 2015 Oct 6;23(10):1815-26.

It is true that in Figure 1—figure supplement 4, the map did not look to contain information at ~3.5 Å resolution (helical grooves absent). The reason is that we used a raw version of the cryoEM map with no further processing (unmasked and unfiltered) and with no enhancement of high frequencies, so these structural details were missing in the rendering. Now, the same figure shows a map with enhanced high resolution features (using a B-factor of -200 1/Å^2^ as stated in Materials and methods). The FSC curve calculated between the cryoEM and the atomic model is now included in Figure 1—figure supplement 3.

4) In the subsection “CP and ssRNA interaction”

*. Interpretation of the CP-complementation studies (e.g., "suggesting a disruption of the CP-RNA interaction") seems to be flawed. Previous studies showed CP mutants that do not interrupt CP-RNA interactions and can still assemble virions (e.g., a C-terminal trucation) are defective in cell-cell movement. CP mutants that do not assemble into virions can complement the defect of cell-to-cell movement. CP mutants that do not assemble into virions can complement the defect in cell-cell movement.*

The straightforward interpretation of these results is that:

"The three amino acids at the binding pocket for nucleotide at position 2 (R124, D163 and K196) are required to complement the cell-to-cell movement of the CP-defective PepMV mutant […]“

As it is first stated in the manuscript (in the revised version). The same sentence continues:

“[…] suggesting a disruption of the CP-RNA interaction needed in virus intercellular transport.”

Since the three amino acids are located in the binding pocket for the RNA, and the cell-to-cell movement needs a protein-RNA complex, it is reasonable to infer such a conclusion. Since we say "suggesting", it does not imply that it has been fully demonstrated, but we think that the interpretation is more than plausible and it needs to be stated. Therefore, we would like to keep it.

5) In the last paragraph of the Introduction and in the first paragraph of the subsection “Structural homology between CP from potexviruses and NP from phleboviruses“

*. The icosahedral shell in RVFV is not a "capsid", which refers instead (in this case) to the helical structure associated with the RNA inside the particle. It is the "icosahedral shell of glycoproteins".*

We agree that terms such as "capsid", "shell" or "coat" can be understood in different ways when describing viral architecture. In the present comparison between potexvirus and phlebovirus the use of these words can be confusing. We have removed "capsid" and now use the suggested "icosahedral shell of glycoproteins", which is more specific and clear.

*6) The title should be changed. The data do not "reveal gene transfer" –*

*such a revelation could come only from an observation of a transfer intermediate. Suggest: The structure of a flexuous-helix plant virus shows homology of its capsid protein with the nucleoprotein of animal bunyaviruses.*

We have changed the title to:

"The near-atomic cryoEM structure of a flexible filamentous plant virus shows homology of its coat protein with nucleoproteins of animal viruses"

*7) The Abstract should be revised along the lines suggested below:*

Flexible filamentous viruses include economically important plant pathogens.Their viral particles contain several hundred copies of a helically arrayed coat protein (CP) protecting a (+)ssRNA.We describe here a structure at 3.9 Å resolution, from electron cryomicroscopy, of Pepino mosaic virus (PepMV), a representative of the genus Potexvirus (family Alphaflexiviridae).Our results allow modeling of the CP and its interactions with viral RNA.The overall fold of PepMV CP resembles that of nucleoprotein (NPs) from the genus Phlebovirus (family Bunyaviridae), a group of enveloped (-)ssRNA viruses.The main difference between potexvirus CP and phlebovirus NP is in their C-terminal extensions, which appear to determine the characteristics of the distinct multimeric assemblies – a flexuous, helical rod or a loose ribonucleoprotein (RNP).The homology suggests gene transfer between eukaryotic (+) and (-) SSRNA viruses.

We have edited the Abstract following the suggestions.